# Hand and Foot Selection in Mental Body Rotations Involves Motor-Cognitive Interactions

**DOI:** 10.3390/brainsci12111500

**Published:** 2022-11-04

**Authors:** Stephan Frederic Dahm, Emiko J. Muraki, Penny M. Pexman

**Affiliations:** 1Department of Psychology, Faculty of Psychology and Sport Science, University of Innsbruck, 6020 Innsbruck, Austria; 2Institute of Psychology, Department of Psychology and Sports Medicine, UMIT TIROL—Private University of Health Sciences and Health Technology, 6060 Hall in Tirol, Austria; 3Department of Psychology, Faculty of Arts, University of Calgary, Calgary, AB T2N 1N4, Canada

**Keywords:** motor imagery, action imagery ability, mental action representations

## Abstract

Action imagery involves the mental representation of an action without overt execution, and can contribute to perspective taking, such as that required for left-right judgments in mental body rotation tasks. It has been shown that perspective (back view, front view), rotational angle (head-up, head-down), and abstractness (abstract, realistic) of the stimulus material influences speed and correctness of the judgement. The present studies investigated whether left-right judgements are more difficult on legs than on arms and whether the type of limb interacts with the other factors. Furthermore, a combined score for speed and accuracy was explored to eliminate possible tradeoffs and to obtain the best possible measure of subjects’ individual ability. Study 1 revealed that the front view is more difficult than the back view because it involves a vertical rotation in perspective taking. Head-down rotations are more difficult than head-up rotations because they involve a horizontal rotation in perspective taking. Furthermore, leg stimuli are more difficult than hand stimuli, particularly in head-down rotations. In Study 2, these findings were replicated in abstract stimuli as well as in realistic stimuli. In addition, perspective taking for realistic stimuli in the back view is easier than realistic stimuli in the front view or abstract stimuli (in both perspectives). We conclude that realistic stimulus material facilitates task comprehension and amplifies the effects of perspective. By replicating previous findings, the linear speed-accuracy score was shown to be a valid measure to capture performance in mental body rotations.

## 1. Introduction

Perspective taking occurs in many situations of daily life. For example, one may withdraw one’s own hand when watching another person putting their hand into a hot flame. Another example is a soccer coach who synchronously imitates a goal-kick of his striker without himself being in position or having a ball. The mechanisms underlying this aspect of everyday behavior are still not fully understood. Therefore, the present studies aimed to replicate and extend investigations on whether perspective taking follows certain rules that are based on action imagery theories [1] and embodied cognition [2]

As a specific type of imagination [3], *action imagery* (also called motor imagery) refers to the imagination of one’s own movement without actually executing the movement [1,4]. In contrast to visual imagery, action imagery may involve the imagination of all types of perception that include motor, kinesthetic, tactile, and visual elements of oneself, of the environment, as well as of the action consequences [5,6]. The visual content can be imagined either from a first-person perspective (i.e., through one’s own eyes during the action) or a third-person perspective (i.e., seeing oneself doing an action through an observer’s eyes) [7]. It is proposed that action imagery draws on mechanisms from action execution [1,5,8] and recruits neural regions similar to those used for action execution [1,9]. In this way action imagery is consistent with theories of embodied cognition, which propose that offline cognitive processes such as memory or language processing engage perceptual and motor neural systems [10,11]. For instance, proprioception of one’s own body posture is more relevant when imagining in the first-person perspective than in a third-person perspective [12], which indicates embodied cognition, e.g., a simulation of one’s own actions and body [8,13]. Similarly, neural activation in somatosensory areas is stronger in the first-person perspective than in the third-person perspective [12].

As in overt action execution, in action imagery, inverse models select the motor commands for the upcoming action which involve information about the corresponding muscle activations. An efference copy of the motor commands is then used by forward models to predict the action consequences [5,14]. In action imagery, inhibition processes [15,16] are assumed to prevent the motor commands from overtly activating the effectors. In the present studies, we investigated the involvement of action imagery in perspective taking in a *mental body rotation task* [17] by showing whole body figures to participants who judged whether the figure raised the left or right limb. Solving mental (body) rotation tasks involves executive functions such as perceptual attention, visual working memory [18], and decision making [19]. Subjects need to quickly select between relevant and irrelevant features of the visual stimuli, keep the updated image stored in working memory in order to mentally rotate it, and then decide about the raised limb [20].

Left-right judgements in a mental body rotation task are presumed to involve perspective taking, i.e., imagining oneself in the position of the human figures [21]. It is assumed that perspective taking involves the same mechanisms that are used in action imagery [4,17,22]. For instance, one may imagine rotating their own body towards a position that is congruent with the presented figure. Hence, this *perspective taking* involves egocentric perspective transformations if the depicted position of the human figure differs from the participant’s position, because the task requires that participants adjust their own reference frame to that of the depicted figure [23,24,25]. The idea that perspective taking involves a covert action of oneself has been supported by corresponding neural activity in motor areas [26] and somatosensory areas [27] during hand rotations. However, an alternative explanation is that perspective taking involves mechanisms required for mental object transformation that do not involve action imagery [17]. For instance, one may imagine rotating the figure towards a position that is congruent with one’s own body position [28]. 

It has been shown that participants who are experts in executing body rotations outperformed control participants in left-right judgments that involve perspective taking [29], but not in same-different judgements that do not involve perspective taking [17,30]. In addition, experts in executing body rotations outperformed control participants only when making judgements about upside-down body positions, but not when making judgements about head-up body positions [17,30]. Furthermore, in experts and novices, response times and error rates increase with angular disparity from head-up positions [17,31,32,33] or with abstractness of the stimuli [30]. For instance, when using photographs rather than line drawing figures and stick figures, response times (RTs) were shorter for the more realistic photograph stimuli than the more abstract stimuli, particularly for experts in executing body rotations [30].

It has previously been shown that efficiency scores combining speed (RTs) and accuracy (error rates: ER) are better suited to show individual differences in laterality judgements [34]. Previous studies have used an inverse efficiency score—RT/(1 − ER)—to combine these measures, however, this score does not take into account differences in variance between the two measures [35,36]. Therefore, *linear speed-accuracy scores* (LISAS) have been proposed as an alternative way to combine speed and accuracy, while also considering their respective variances. The LISA score is practically identical to the response time in case a participant made no errors. However, the LISA score increases with an increase of committed errors. Hence, good performance is indicated by low LISA scores which indicate fast and correct responses. In comparison to analyses of RTs alone [37] or to separate analyses of RT and errors [17,30,31,32], the LISA score additionally tracks performance increments that are caused by (lacking) accuracy, as with a speed-accuracy trade-off [17,30,31].

In the present studies, our goal was to replicate previous findings of perspective and rotation effects on mental body rotation task performance using LISA scores. Hence, by investigating healthy participants and using a counterbalanced within-group experimental design, we expected better performance (lower LISA scores) in head-up positions than in head-down positions [17,32,33], and in a back view/egocentric perspective rather than in a front view/allocentric perspective [17]. Such effects of positional (in)congruency [38] between one’s own body position and the depicted figure could indicate the involvement of action imagery in mental body rotations.

In addition to the replications of these previously observed effects, we used stimuli that included arm and leg items. Leg items have not previously been investigated in whole body laterality judgement tasks. By using leg items, we intended to increase task difficulty of the laterality judgements. In daily situations, distinguishing between the left and right leg is less common than distinguishing between the left and right arm. Furthermore, the lower limbs may be more sensitive for some motor control functions than the upper limbs [39]. In accordance with this assumption, RTs have been observed to be shorter in hand items than in foot items [37]. Therefore, we expected better performance (lower LISA scores) for raised arms than for raised legs. Effects of the limbs would provide additional evidence for the use of action imagery in mental body rotation tasks.

In the second study, we compared realistic stimuli (human-like, gender-free avatars) with the abstract stimuli used in the first study (elliptical drawings of a human body). Realistic stimuli are assumed to make the task more intuitive [30]. Using LISAS, we expected to replicate better performance in realistic than in abstract stimuli [30] not only for arm, but also for leg items. 

## 2. Study 1

For the first study, we created abstract stimulus material with elliptical drawings of a human body. The aim of Study 1 was to replicate findings of perspective and rotational angles on linear speed-accuracy scores. Further, we manipulated the raised limb, expecting higher scores on raised legs than on raised arms.

### 2.1. Methods of Study 1

#### 2.1.1. Participants of Study 1

The link for participation was provided to interested students at the University of Calgary. Participants were not familiarized with the task prior to the study. Because the data collection was online where distractions cannot be controlled for, a rigorous outlier analysis was performed on RTs and error rates. Of 182 English-speaking participants, 27 participants were excluded from analysis because they were just clicking through (*n* = 9; error rates close to 50% and RTs below 500 ms), they were inattentive or distracted (*n* = 7; error rates close to 50%), they did not comply with the instructions (*n* = 9; error rates close to 100% in front view, but not in back view), due to missing values in RT of correct responses (*n* = 2) or due to LISA scores above 3 *SD* from the mean (*n* = 1). The remaining 153 participants (128 females, 24 males, 1 else) were on average 21 years old (*SD* = 4.5, range from 18 to 44 years). The laterality index (assessed with the Edinburgh Handedness Inventory, [40]) ranged from −100 to +100 with the mean (*M* = 73.9, *SD* = 37) indicating mainly right-handers. The Vividness of Movement Imagery Questionnaire 2 VMIQ-2; [7] indicated that participants had clear and vivid external visual imagery (*M* ± *SD* = 2.2 ± 0.9), internal visual imagery (*M* ± *SD* = 1.8 ± 0.8), and kinesthetic imagery (*M* ± *SD* = 2 ± 0.9). All participants gave informed consent. The study was in accordance with the Belmont Report [41] and was approved by the local ethics committee.

The data reported here were collected in a larger experiment involving language processing tasks that are not the focus of the present manuscript [42], and the sample size was selected based on having sufficient power to detect effects in the language tasks. For the purposes of the present analysis, we conducted a post-hoc power analysis for an interaction between eight conditions (the combination of perspective, rotation and limb) using G*Power [43]. The LISA score was the primary outcome measure. We assumed an effect size of *f* = 0.25 and alpha was set at 0.05, which resulted in the power (1-beta) of 0.87. 

#### 2.1.2. The Stimulus Material of Study 1

We created gender-free items that showed an abstract depiction of a human figure (Figure 1) either in front view (i.e., the figure is facing the participant) or in back view (i.e., the figure is looking in the same direction as the participant). The eight rotational angles were 0, 45, 90, 135, 180, 225, 270, and 315. The figure raised one limb (left arm, right arm, left leg, right leg). This resulted in a total of 64 stimuli. 

#### 2.1.3. Task and Procedure of Study 1

The experiment was created using the PsychoPy Builder interface [44] and run online using PsychoJS Version 3.2 [45]. Participants were told that the visual stimuli represent the figure of a human body which will be rotated. They were asked to press the ‘D’ key if the raised limb (foot or arm) is left and to press the ‘K’ key if the raised limb is right. Participants were told that in the front view they could see the face of the figure, whereas in the back view no face was visible. They performed eight familiarization trials with stimuli featuring every combination of perspective (i.e., front or back view), limb (i.e., arm or leg) and side (i.e., left or right), all presented at 0 rotation. This was followed by a block of 64 trials in which the stimuli (or conditions) were presented in random order. Participants’ responses triggered the presentation of the next stimulus. 

#### 2.1.4. Data Analysis of Study 1

RT was defined as the interval between presentation of the stimulus and participants’ response. The error rate indicates the percentage of incorrect responses. Median RTs and the percentage of errors were calculated for each perspective, rotation, and limb. Rotational angles were differentiated between head-up (−45, 0, +45) and head-down (−135, 180, 135), which have been shown to make the largest difference to behavioral responses [30]. Analysis of RTs and error rates revealed that the main effects could be observed in both measures (see Appendix A). To take into account the speed-accuracy tradeoff in individuals, linear integrated speed-accuracy scores (LISAS) were calculated [35]. Because RTs are usually not normally distributed, we used the median and median absolute deviation (MAD; instead of the mean and standard deviation) when calculating these scores. Hence, either the median RT of correct responses (if the error rate is 0%) or median RT of correct responses + error rate × *MAD* (RT)/*SD* (ER) was used. For the repeated measures ANOVA, partial eta squared (η*_p_²)* is reported as effect size. Further comparisons were conducted using t-tests with Holm adjusted pairwise comparisons with Cohen’s *d* as effect size. Where appropriate, minimum (*p*_min_) or maximum (*p*_max_) statistical values are reported. For all analyses, the probability of errors of the first kind was set at α = 0.05. The tidyverse 1.3.1 [46] and rstatix 0.7.0 [47] packages were used for analysis using version 1.2.5033 of RStudio [48] and the ggplot2 package for the generation of graphs [49]. Data as well as the syntax for data analyses are available at https://osf.io/ymf8w.

### 2.2. Results of Study 1

A repeated measures ANOVA with the within-subject factors of perspective (front view, back view), rotation (head-up, head-down), and limb (arm, leg) was calculated on the LISAS. Means and standard errors of LISAS are shown in Figure 2. 

The significant main effect of *perspective*, *F* (1, 152) = 121.7, *p* < 0.001, η*_p_²* = 0.45, indicated significantly higher scores in the front view (*M* ± *SD* = 2.1 ± 1.1) than the back view (*M* ± *SD* = 1.7 ± 0.9). The significant main effect of *rotation*, *F* (1, 152) = 360.7, *p* < 0.001, η*_p_²* = 0.7, indicated significantly higher scores in head-down rotations (*M* ± *SD* = 2.4 ± 1.1) than head-up rotations (*M* ± *SD* = 1.4 ± 0.6). The significant main effect of *limb*, *F* (1, 152) = 95, *p* < 0.001, η*_p_²* = 0.39, indicated significantly higher scores for leg items (*M* ± *SD* = 2 ± 1) than arm items (*M* ± *SD* = 1.8 ± 0.9). The significant interaction between *rotation and limb*, *F* (1, 152) = 13.6, *p* < 0.001, η*_p_²* = 0.08, indicated that the difference between limbs was significantly larger in head-down rotations (Δ*M* = 0.3) than in head-up rotations (Δ*M* = 0.1; *p* < 0.001, *d* = 0.3). All remaining interactions were not significant, η*_p_²* < 0.01.

### 2.3. Discussion of Study 1

Visual inspection of the effect sizes of RT, ER (in the Appendix A) and LISAS indicated that effect sizes in RT and ER were lower than in the LISA scores. Hence, LISA scores reflect the performance better than single analyses of RT or ER, as the effects in both measures and the relative values considering the variances of the two measures are taken into account in the LISAS. For instance, one participant may slow down in RTs to keep the ER low in more difficult items. In contrast, another participant may accept an increase in ER to keep RTs at the same level in more difficult items. In case of speed-accuracy tradeoffs in individual participants, the effects in RT and ER can also neutralize each other in the LISAS, making it a more appropriate measure, particularly to compare between individuals who differ in speed-accuracy preferences. This could also explain the small correlations between ER and RT (as shown in the Appendix A).

As observed previously [17], performance was better in the back view than in the front view. It is assumed that this results from participants imagining a *perspective rotation* on the vertical axis, which is necessary in the front view, but not in the back view. In the back view, the participant’s body is already congruent with the body position of the figure of the stimulus. Hence, when putting oneself in the perspective of the targeted stimulus figure to accurately make the left-right judgement, one uses imagery in a way that is consistent with the principles of embodied cognition [2,10].

Like the perspective factor, we also replicated the *rotation effects* observed in previous studies [17,32,33]. Performance was better in head-up rotations (rotations of maximal 45 degrees to the left or right) than in head-down rotations (rotations between 135 and 225 degrees). Similar to the perspective factor, it is assumed that this results from an imagined rotation. Such rotations on the horizontal axis are more time consuming and produce more errors proportional to the rotational angle [17,31,32,33]. In the present study, LISAS were higher in head-down rotations than in head-up rotations. Head-up stimuli do not involve such time-consuming and error-prone imagined rotations, which explains the better performance compared to head-down stimuli. This applies even for those head-up stimuli that involve slight rotational angles of 45 degrees.

As expected, the raised *limb* had an additional influence on mental body rotations. This was observed particularly in difficult rotational angles when the head is down. Performance was lower in leg stimuli than in arm stimuli, which may emerge due to various reasons. First, in daily life, we use our legs less often than our arms [39]. This applies particularly for students when they are sitting in the classroom or office. Second, distinguishing between left and right is often less important for legs than it is for arms. For instance, the right hand is typically used for a handshake, whereas there is no such greeting method or other everyday behavior for legs that favors left or right. Third, congruency with the response action [38] may have increased these effects, as participants responded with the fingers of the hands and not the feet. Hence, responding with the fingers (of the hand) to hand stimuli was more congruent than responding with the fingers (of the hand) to feet stimuli. In any event, differentiating between left and right legs increased task difficulty, which may be useful to measure individuals’ action imagery ability more precisely [4].

## 3. Study 2

In Study 2, our goal was to replicate and extend the findings of Study 1. Therefore, we expected higher LISA scores in raised legs than in raised arms. Furthermore, we used more realistic stimuli (human-like avatars) to compare them with the abstract stimuli of Study 1. Using LISA scores, we expected better performance for realistic stimuli than for abstract stimuli [30]. 

As a secondary research aim, we assessed participants’ general self-efficacy [50] and a German version [51] of vividness in action imagery (VMIQ-2, [7]) to test discriminative and convergent validity. As observed previously [52], we expected low to moderate (0.2 < *r* < 0.5) correlations between these self-assessment questionnaires (self-efficacy and VMIQ-2). Most importantly, we expected low correlations between self-efficacy and LISA scores of the mental body rotations, as they assess different constructs using different methods. In contrast, action imagery self-assessment questionnaires may measure the same construct as objective measures such as the mental body rotation task. Therefore, we expected low to moderate correlations between the VMIQ-2 ratings and LISA scores.

### 3.1. Methods of Study 2

#### 3.1.1. Participants of Study 2

The link for participation was disseminated by student project members to their friends and to interested students at the UMIT—the Tyrolean Private University. As in Study 1, a rigorous outlier analysis was performed on RTs and error rates. Of 146 German-speaking participants, 24 participants were excluded from analysis because they were just clicking through (*n* = 6; error rates close to 50% and RTs below 500 ms), they were inattentive or distracted (*n* = 2; error rates close to 50% and large variance in RTs), they did not comply with the instructions in abstract stimuli (*n* = 9; error rates close to 100% in back view, but not in front view), or due to extreme outliers in RTs (above 5 s; *n* = 3) or error rates (above 50% in several but not all conditions; *n* = 4). The remaining 122 participants (72 females, 50 males) were on average 28.5 years old (*SD* = 10.3, range from 18 to 71 years) and mainly right-handed, as self-reported by the participants (*N* = 109). A German version [51,52] of the VMIQ-2 [7] indicated that participants had clear and vivid external visual imagery (*M* ± *SD* = 2.1 ± 0.7), internal visual imagery (*M* ± *SD* = 1.8 ± 0.7), and kinesthetic imagery (*M* ± *SD* = 1.9 ± 0.7). All participants gave informed consent. The study was in accordance with the Belmont Report [41] and was approved by the local ethics committee.

The required sample size for an interaction between 16 conditions (the combination of abstractness, perspective, rotation and limb) was estimated with G*Power [43]. The primary outcome measure used for sample size estimation was the LISA score. We assumed an effect size of *f* = 0.25. Alpha was set at 0.05 and the power (1-beta) at 0.8 which resulted in a minimum sample size of *N* = 128. Because the estimated sample size was not achieved in the final sample (*N* = 122), the power for medium effects (*f* = 0.25) was 0.78, which is only slightly below the recommended value of 0.8 [53].

#### 3.1.2. The Stimulus Material of Study 2

In addition to the stimuli from Study 1, gender-free realistic avatars were created for purposes of Study 2, either in front view, i.e., the figure is facing the participant, or in the back view, i.e., the figure is looking in the same direction as the participant (Figure 3). The stimuli were created using makehuman in blender [54] by selecting 50% male and 50% female characteristics in body and face.

#### 3.1.3. Task and Procedure of Study 2

The experiment was run online using OpenSesameWeb Version 3.3.11 [55] and JATOS [56]. The experiment file is available at https://osf.io/ymf8w. As in Study 1, participants were instructed that the visual stimuli represent the figure of a human body that will be rotated. They were asked to press the ‘X’ key if the raised limb (foot or arm) is left and to press the ‘Y’ key if the raised limb is right. They were told that in the front view they could see the face of the figure, whereas in the back view no face was visible. To avoid learning effects during the assessment, participants started with four randomized familiarization blocks of 64 trials where either one of abstractness (abstract vs. realistic) or the limbs (arm vs. leg) was fixed for the block (e.g., a block of only abstract stimuli). This was followed by the assessment block of 128 trials in which all stimuli (or conditions) were presented in random order. A fixation dot was presented for 500 ms before each stimulus appeared on the screen. Participants’ response triggered the presentation of the next fixation dot.

#### 3.1.4. Data Analysis of Study 2

As in Study 1, linear integrated speed-accuracy scores (LISAS) were calculated and analyzed. Additionally, correlations between the vividness of movement imagery [51], self-efficacy [50], and the LISAS were calculated. Data as well as the syntax for data analyses are available at https://osf.io/ymf8w.

### 3.2. Results of Study 2

A repeated measures ANOVA with the within-subject factors of abstractness (abstract, realistic), perspective (front view, back view), rotation (head-up, head-down), and limb (arm, leg) was calculated on the LISAS. Means and standard errors of LISAS are shown in Figure 4. Results of the ANOVA are shown in Table 1.

The significant main effect of *perspective* was modified by the significant interaction between abstractness and perspective. In realistic stimuli, the scores were significantly higher in the front view (*M* ± *SD* = 1.5 ± 0.9) than the back view (*M* ± *SD* = 1.2 ± 0.7, *p* < 0.001, *d* = 0.71). In abstract stimuli, the scores did not significantly differ between front view (*M* ± *SD* = 1.4 ± 0.7) and back view (*M* ± *SD* = 1.4 ± 0.7, *p* = 0.287, *d* = 0.1).

The significant main effect of *rotation* indicated significantly higher scores for head-down rotations (*M* ± *SD* = 1.5 ± 0.8) than head-up rotations (*M* ± *SD* = 1.2 ± 0.7). The significant interaction between rotation and perspective indicated that this difference was significantly larger in the back view (Δ*M* = 0.5) than in the front view (Δ*M* = 1.3, *p* < 0.001, *d* = 0.74).

The significant main effect of *limb* indicated significantly higher scores for leg stimuli (*M* ± *SD* = 14.7 ± 0.8) than arm stimuli (*M* ± *SD* = 1.2 ± 0.6). The significant interaction between abstractness and limb indicated that this difference was significantly larger for realistic stimuli (Δ*M* = 3.6) than abstract stimuli (Δ*M* = 1.1, *p* < 0.001, *d* = 0.5). 

The significant main effect of *abstractness* was modified by the significant interaction between abstractness, perspective, and limb. For arm stimuli, the scores were significantly higher for abstract stimuli than realistic stimuli in the front view (Δ*M* = 0.11, *p* < 0.001, *d* = 0.32) and back view (Δ*M* = 0.26, *p* < 0.001, *d* = 0.69). In contrast, for leg stimuli, the scores were significantly lower for abstract stimuli than realistic stimuli in the front view (Δ*M* = −0.23, *p* < 0.001, *d* = −0.39), but not in the back view (Δ*M* = 0.09, *p* = 0.067, *d* = 0.17).

Pearson correlations between self-efficacy (SE, [49]) vividness of external visual (EVI), internal visual (IVI), and kinesthetic (KIN) imagery [7] and linear integrated speed-accuracy scores are shown in Figure 5. High interfactor-correlations were observed between the subdimensions of the VMIQ-2 [7]. Moderate correlations were observed between self-efficacy and vividness of action imagery. High correlations were observed between all LISA scores of the mental body rotations. Low correlations were observed between the questionnaires and the LISA scores.

In addition, we performed an *exploratory analysis* by calculating difference scores between the baseline condition without any rotational movement in the back view with heads up and the other conditions (e.g., front view head-down realistic legs—back view head-up realistic legs) to separate the unique variance of action imagery from other possible constructs in the mental body rotation task. Correlations of these difference scores are also shown in Figure 3.

### 3.3. Discussion of Study 2

As in Study 1, we observed higher scores for head-down rotations than head-up rotations, which is consistent with the findings of several previous studies [17,31,32,33]. In Study 2, this effect was even amplified in the back view, which allows a simple egocentric perspective without the need of a vertical rotation necessary for head-up rotations in the front view. Additionally, we observed higher scores for the front view than the back view [17]. However, this was observed for realistic but not abstract stimuli. The data pattern suggests that the egocentric perspective in realistic stimuli facilitates perspective taking, resulting in lower LISAS. An explanation for these findings may be that head-up figures in the back view do not require any mental rotations, only perspective taking, which is then facilitated if the stimuli are more realistic. Head-down figures in the back view require lateral horizontal mental rotations of one’s own body to fully take the perspective and make a left/right judgement. Therefore, there was a large difference between the rotations in the back view. For the front view, however, both rotational angles (head-up and head-down) require a mental rotation. Head-up figures in the front view require a vertical mental rotation. One may assume that in head-down figures in the front view the vertical and lateral horizontal mental rotations are performed in a stepwise manner [33] which should have increased the scores considerably. However, this was not observed. Therefore, we assume that participants performed a frontal horizontal rotation (like a back-flip or front-flip), which was still slightly more difficult (or time-consuming) than a single vertical rotation, but easier (less time-consuming) than performing the rotations in a stepwise manner.

In accordance with Study 1, we observed higher scores for leg stimuli than arm stimuli. This was even amplified in realistic stimuli. One reason for this could be that the angle of the *limb* was more difficult to perceive in realistic stimuli than in abstract stimuli. Participants’ verbal reports indicated that for realistic stimuli they needed to focus more strongly on both the foot the figure was standing on and the foot that was raised. It is possible that this was not the case in abstract stimuli where they focused on the raised limb only. From an action imagery point of view, this may indicate that for realistic stimuli, participants intended to replicate the balancing position on the standing foot to then imagine raising the other foot. This suggests that participants engaged in simulations of somatosensory experience to a greater degree for foot items, particularly for realistic stimuli. These assumptions are in line with the observation of lower scores in realistic stimuli than in abstract stimuli when arms were raised, but not when legs were raised. When legs were raised this was reversed in the front view. Hence, the allocentric perspective made the judgements on realistic leg figures more difficult.

The correlations in the present study showed that action imagery self-assessments via questionnaire and experimental assessments do not correlate. Similarly, such assessments do not correlate in visual imagery [3,57]. For instance, mental rotations of characters and numbers [58] did not correlate with the vividness of the visual imagery questionnaire [59]. In action imagery, it has been argued that the lack of correlations between experimental and self-report assessments [60] is due to the *methods of measurement* capturing different components of action imagery. However, such findings may also indicate that the different methods may capture constructs other than action imagery, because experimental methods always involve a combination of several abilities [3]. For instance, mental body rotations may not only require action imagery ability, but also independent constructs such as left-right disorientation, and closely related constructs such as fluid intelligence or working memory capacity. To rule out this argument, we calculated explorative difference scores that partialled out other constructs by using the position without any rotations (back view head-up) of the mental body rotation task as baseline. However, like the absolute scores, these difference scores did not correlate with the self-ratings of the VMIQ-2. Finally, the lack of correlation between self-ratings and objective measures in action imagery may be due to biases in self-reports [4,61] and a lack of variance in self-report questionnaires [52].

Interestingly, the difference scores were more strongly correlated with the absolute head-down scores than with the absolute head-up scores. This shows that the variance in the difference scores is mainly influenced by the rotation on the horizontal axis (which indicates the imagined action) and not as much by other constructs that may influence response times (e.g., general response times, working memory capacity). Furthermore, variance was also influenced by the perspective (rotations on the vertical axis), although not as strong as by rotations on the horizontal axis.

## 4. General Discussion

The results of both studies provide a replication of previous mental body rotation studies [17,31,32], showing that the rotational angle strongly affects participants’ responses, not only in RTs but also in a combined measure of RT and ER. The LISA scores were higher (indicating worse performance) in head-down positions than in head-up positions. Furthermore, the scores were higher in the allocentric front view than in the egocentric back view [17], indicating that participants engaged in action imagery [1] for the task decisions, consistent with theories of embodied cognition [11]. Moreover, the scores were higher for abstract stimuli than realistic stimuli [30]. In addition to these replications, the present studies showed that leg judgments were more difficult to solve than arm judgements. 

Regarding the *vertical and horizontal rotations*, it can be argued that this does not necessarily imply action imagery, i.e., a mental representation of one’s own action in the absence of completing that action. Imagery may involve either a mental rotation of oneself or a mental rotation of the stimulus to gain congruency between one’s own perspective and the depicted perspective [28]. However, it has been shown that hand judgments (as used in the present study) involve representations grounded in the motor system, while same-different judgements involve object-based representations grounded in the visual system [32]. 

Additionally, it can be argued that action imagery usually involves imagination of oneself, whereas the stimuli are neutral. However, verbal reports of participants indicated that they tended to imagine rotations of their own body to put themselves into the position of the figure. Furthermore, it has been shown that there is no advantage if the figures in a mental body rotation task show pictures of oneself instead of another person [32,38]. This implies that participants adopt a strategy to mentally rotate their own body to be congruent with a depicted figure, regardless of whether they implicitly associate themselves with the figure or not. Such first person imagery is most likely enriched by embodied kinesthetic action imagery where proprioceptive information is more relevant than in visual action imagery [12].

When comparing the absolute values of both studies (see Figure 2 and Figure 4), it becomes apparent that the median LISAS in Study 1 (from 1.25 to 2.5) are at least tendentially higher than in Study 2 (from 1 to 1.5). Most likely, this resulted from the additional familiarization blocks in Study 2 (4 × 64 trials) compared to just eight trials in Study 1. Additional analyses in Study 2 (see Appendix A) indicated significant improvements over the familiarization blocks. Hence, repeated familiarization with the task leads to lower LISAS.

### Limitations and Perspectives

The *angle of the raised limb* may affect both speed and accuracy of the judgment in mental body rotations. Unfortunately, this effect was not consistent between abstract and realistic images, nor between arms and legs. Future studies may wish to investigate whether the angle between the raised limb and the body has an impact on mental body rotation scores.

It remains unclear whether the mental body rotation task involves internal visual imagery, external visual imagery, or kinesthetic imagery. None of these dimensions of the VMIQ-2 [7] correlated with the mental body rotation scores. Still, it remains likely that at least one of these dimensions are related to action imagery ability that is used during mental body rotations [4]. Future studies may focus on participants’ *strength of representation* during mental body rotations. For instance, after mental body rotations, participants could be asked to report how strongly they focused on these modalities (i.e., visual or kinesthetic) and perspectives (i.e., internal or external) using a provided scale [62].

Our findings have implications for the assessment and measurement of *action imagery ability*. For a more sensitive measure of action imagery ability, we suggest using both leg and arm stimuli to increase the overall difficulty of the task. Furthermore, we suggest using realistic stimuli rather than abstract stimuli, which may prevent misunderstandings of the perspective in some participants (see participant exclusions in both studies). To render out confounding factors in the measure of action imagery ability (such as other cognitive abilities like working memory), we recommend taking the front view head-up condition as a baseline measure that does not require any movement (only perspective taking, but no rotation of one’s own body) to calculate difference measures. 

The present study was not designed to test the effects in different populations. Hence, to provide greater generalizability of the findings, future studies may investigate selected subpopulations. For instance, action imagery ability may differ between gender [63] or movement expertise [63]. Furthermore, patients with pain (or amputations) specific to either legs or arms could be investigated.

From an applied point of view, improved measurements of action imagery ability may be helpful in various fields such as physiotherapy [64], neurorehabilitation [65,66], sports [67], speech and language therapy [68], and music [69]. In action imagery practice (or mental practice) which designates the repetitive use of action imagery to improve motor performance, it has been proposed that high action imagery abilities boost the practice effects [70,71]. Using an objective measure of action imagery ability, individualized interventions may compensate for potential imagery deficits [72].

## 5. Conclusions

The results support the assumption that action imagery is involved in solving the mental body rotation task with left-right judgements [17,31]. An increase in LISA scores was caused by vertical and horizontal rotations that required the imagination of a rotational action. Although it appears likely that action imagery is involved in mental rotations, the object of rotation during action imagery remains unresolved. One does not need to necessarily imagine rotating oneself to be congruent with a depicted figure. In the imagination, the depicted figure could also be rotated. The latter would be a mental object transformation rather than action imagery. However, mental object transformations should not be influenced by the abstractness of the stimuli. Therefore, the increase in LISA scores in abstract stimuli compared to realistic stimuli strongly supports the assumption of action imagery processes in mental body rotations. Such imagery processes may be similar to processes engaged for action execution [1,17], thereby providing further support for embodied theories of cognition [2].

## Figures and Tables

**Figure 1 brainsci-12-01500-f001:**
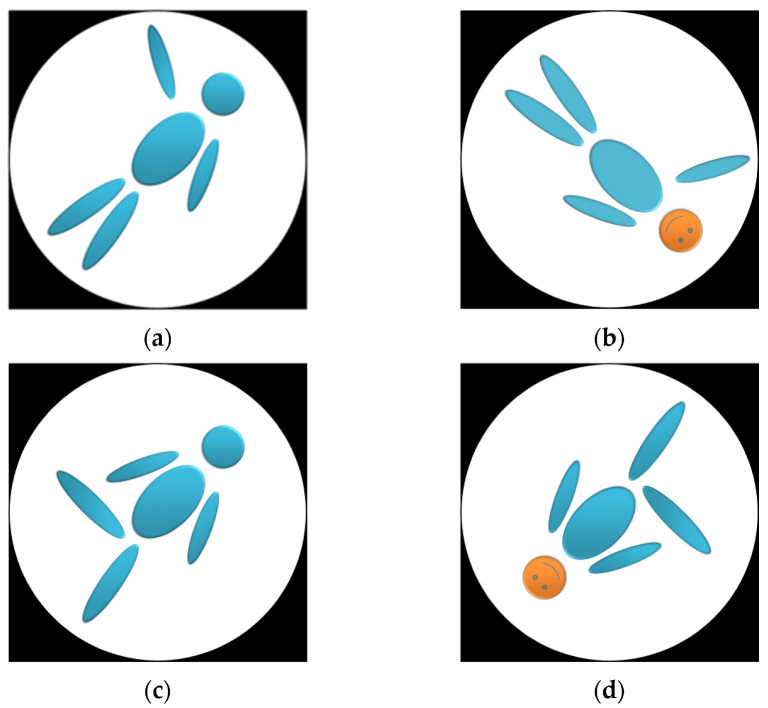
Depiction of the abstract stimuli in Study 1. (**a**) Back view of the left arm raised. (**b**) Front view of the right arm raised. (**c**) Back view of the left leg raised. (**d**) Front view of the right foot raised.

**Figure 2 brainsci-12-01500-f002:**
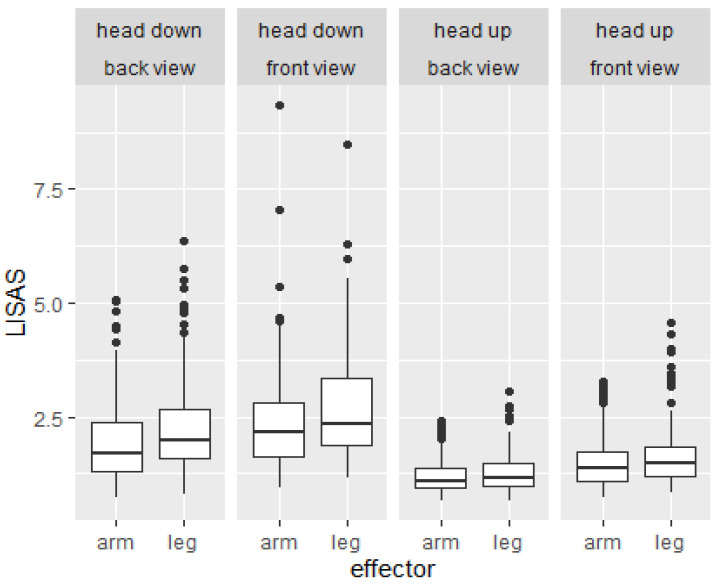
Boxplots of the linear integrated speed-accuracy scores (LISAS) depending on rotation (head-down, head-up), perspective (back view, front view), and limb (arm, leg) in Study 1.

**Figure 3 brainsci-12-01500-f003:**
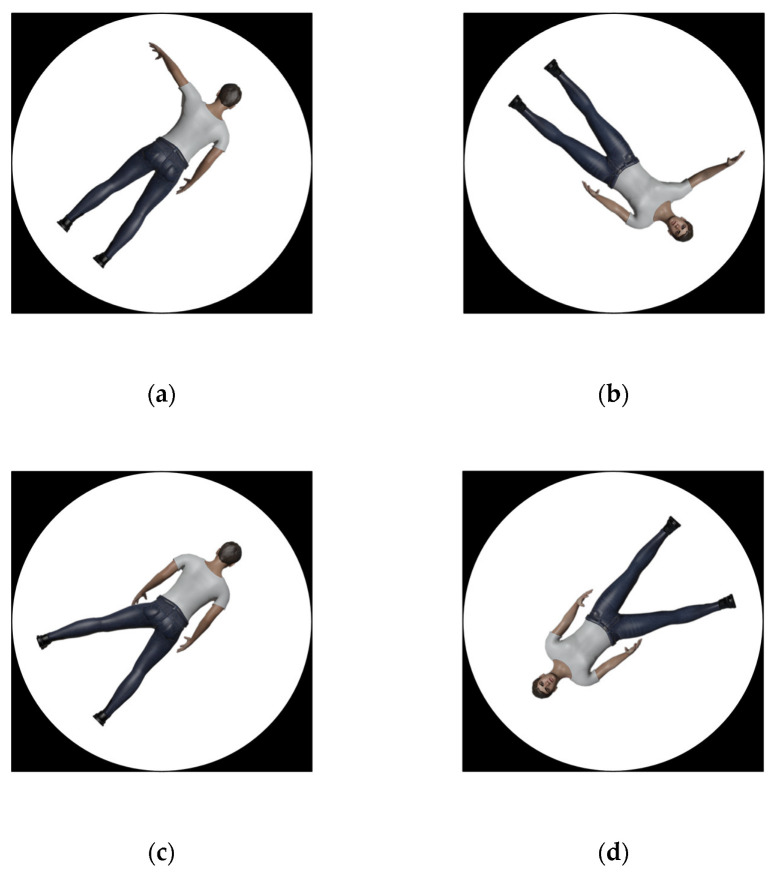
Depiction of the realistic stimuli in Study 2. (**a**) Back view of the left arm raised. (**b**) Front view of the right arm raised. (**c**) Back view of the left leg raised. (**d**) Front view of the right foot raised.

**Figure 4 brainsci-12-01500-f004:**
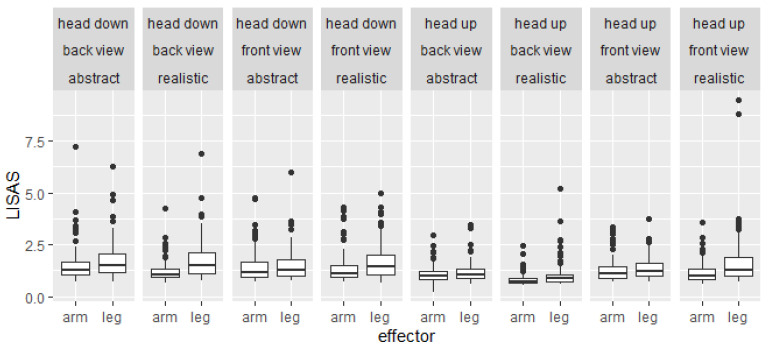
Boxplots of the linear integrated speed-accuracy scores (LISAS) depending on perspective (front view, back view), rotation (head-up, head-down), abstractness (realistic, abstract), and limb (arm, leg) in Study 2.

**Figure 5 brainsci-12-01500-f005:**
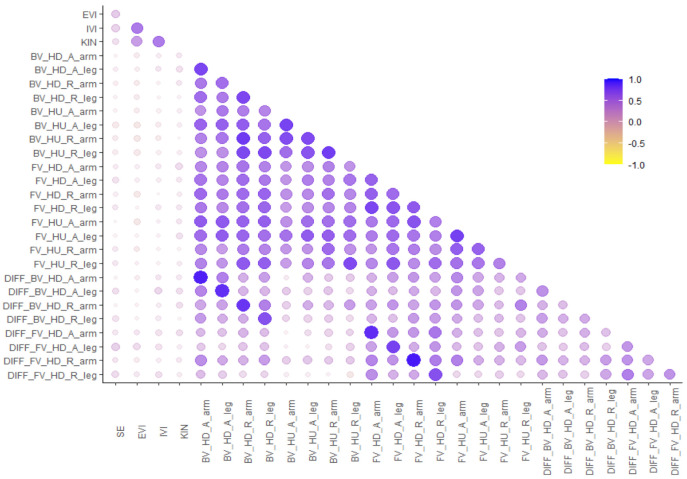
Pearson correlations between self-efficacy (SE; [49]) vividness of external visual (EVI), internal visual (IVI), and kinesthetic (KIN) imagery [7] and linear integrated speed-accuracy scores depending on perspective (front view: FV, back view: BV), rotation (head-up: HU, head-down: HD), abstractness (realistic: R, abstract: A), and limb (arm, leg). Additionally, correlations of the difference score (DIFF) of each condition with the baseline (back view head-up) are shown. Larger and darker circles indicate larger correlations.

**Table 1 brainsci-12-01500-t001:** Statistical values of the ANOVA (*df*_1_ = 1, *df*_2_ = 121) on LISAS in Study 2.

	*F*	*p*	η*_p_²*
Abstractness	5.2	0.025	0.41
Perspective	29.4	<0.001	0.20
Rotation	137.2	<0.001	0.53
Limb	93.0	<0.001	0.44
Abstractness × Perspective	42.0	<0.001	0.26
Abstractness × Rotation	0.1	0.803	<0.01
Abstractness × Limb	30.6	<0.001	0.20
Perspective × Rotation	67.3	<0.001	0.36
Perspective × Limb	0.2	0.643	<0.01
Rotation × Limb	2.9	0.094	0.02
Abstractness × Perspective × Rotation	0.3	0.565	<0.01
Abstractness × Perspective × Limb	5.8	0.018	0.05
Abstractness × Rotation × Limb	0.3	0.564	<0.01
Perspective × Rotation × Limb	22.3	<0.001	0.16
Abstractness × Perspective × Rotation × Limb	3.5	0.065	0.03

## Data Availability

The authors confirm that the data supporting the finding of this study are available within the article and its Appendix A. Data exclusions, manipulations, and all measures in the study are reported in the manuscript. Stimulus material and data are available in the following link: https://osf.io/ymf8w.

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
