# Peer review of "Hand and Foot Selection in Mental Body Rotations Involves Motor-Cognitive Interactions"

_brainsci, 2022, doi:10.3390/brainsci12111500_

Round 1

Reviewer 1 Report

Title: Hand and foot selection in mental body rotations involves motor-cognitive interactions

Article Type: Original article

Summary

In this study, the authors have examined the perspective taking and rotation effects (head-up versus head-down positions, back view versus front view, used stimuli that included arm versus leg items, realistic stimuli versus abstract stimuli) on mental body rotation task performance using LISA scores. The results indicated that the front view is more difficult than the back view, head-down rotations are more difficult than head-up rotations, leg stimuli are more difficult than hand stimuli. The results also showed that perspective taking for realistic stimuli in the back view is easier than realistic stimuli in the front view or abstract stimuli.

General Comment: The manuscript was good and interesting in terms of content and the novelty of the topic and was well written. The manuscript seems to be suitable for publication in the Journal and after correcting a few small cases, it has the conditions to be published in the Journal.

Introduction: Although the introduction talks well about the variables but does not talk more about the theories and its relations like embodied cognition, so it is suggested to talk more and in more detail in this regard.

Method: Please talk more about how to choose the sample size and also the type of research method. Why didn't you use sample size software like Gpower? Please add the exclusion/inclusion criteria in this section. Does your study have an ethical code? Please add it in this section. Please discuss about the participants' level of familiarity as well as skill level with the task.

Results and discussion: The results, discussion and conclusion were written well. However, It is still suggested that the theories mentioned in the introduction be used more and better to justify the results in this section.

Author Response

response letter is attached

Reviewer 2 Report

Thank you for giving me the opportunity to review this manuscript.

This studies investigated  whether left-right judgements are more difficult on legs than on arms and whether the type of limb interacts with the other factors.  The front view is more difficult than the back view, and head-down rotations are more difficult than head-up rotations. Further, leg stimuli are more difficult than hand stimuli.  Realistic stimulus material facilitates task comprehension and amplifies the effects of perspective. Furthermore, the advantage of this article is that the raw data were available.

However, I think it is necessary to revise the manuscript.

1) Please describe the study designs of study 1 and 2 more clearly by using the PECO format (the Participant, the exposure, the control and the outcome). Were the studies cross-sectional study? Please describe the inclusion and exclusion criteria more clearly. Were participants with schizophrenia, intellectual disability, ADHD, or autism spectrum disorder included?? Were participants who took any psychotropic medications included?? 

2) Please describe baseline characteristics of participants in the study 1 and study 2. Please describe any potential confounders and effect modifiers in the study 1 and Study 2. If both studies had the exposure and the control arm, please describe the baseline characteristics in the both arms. For example, how much did the age, gender, baseline assemment scores, and medications affect the outcome?? How was the sample size arrived?

3)  Please describe any statistical methods to control for confounders. In this study some participants were exluded. Then, how were the missing data of these participants handled?

I think it is necessary to revise the manuscript.

Author Response

response letter is attached

Reviewer 3 Report

The article deals with an interesting topic, it is written clearly and briefly. From a technical point of view, everything looks competent and clear. From the point of view of content the article is written too narrowly.

Considering the title and the subject matter of the journal, I believe that the article needs the following revisions.

From my point of view, the article greatly lacks a physiological basis for the phenomena under study. What are the brain mechanisms of mental rotations? What and to what extent are the executive functions involved here? How can the results of study 1 and study 2 be compared considering this?

Also, the introduction needs to address the basics of spatial representations from a psychological, physiological, and psychophysiological perspective.

When the introduction is expanded, the authors can enrich the discussion as well.

Also, the paper should clearly spell out the areas of application of the findings in practice.

Author Response

response letter is attached

Round 2

Reviewer 2 Report

Thank you for giving me the opportunity to review the revised manuscript.

I think it is still necessary to revise the manuscript.

1) You explained that "Using a within-subject design has the advantage that there are no differences in baseline data between experimental conditions, thereby reducing the risk for potential confounders compared to between-subject designs." This explanation is totally wrong. It is  necessary to assess the baseline characteristics even in well-designed randomized controlled trials. In particular, Experiment 2 did not arrived at the estimated sample size finally. Furthermore, randomization procedures were not clearly described. Furthermore, no primary outcome was described, which may lead to type 1 and 2 errors. These factors may increase the risk of bias in this study. Please clearly describe baseline characterstics in the Experiment 1 and 2.

2) Please describe that the sample size was smalled than the estimated one in the Experimental 2, and the populations may not be blanced enough in the Experiment 1 and 2, in the limitation section.

I am sorry to inform you that it is still important to revise the manuscript.

Author Response

see document

Reviewer 3 Report

Dear authors, thanks for your reply. You have worked with all my comments.

Author Response

see document
